# Practical Real Time Recurrent Learning with a Sparse Approximation to the Jacobian

**Jacob Menick**[*]
DeepMind
University College London
jmenick@google.com

**Erich Elsen**[*]
DeepMind
eriche@google.com

**Utku Evci**
Google

**Simon Osindero**
DeepMind

**Karen Simonyan**
DeepMind

**Alex Graves**
DeepMind

## Abstract

Recurrent neural networks are usually trained with backpropagation through time, which requires storing a complete history of network states, and prohibits updating the weights 'online' (after every timestep). Real Time Recurrent Learning (RTRL) eliminates the need for history storage and allows for online weight updates, but does so at the expense of computational costs that are quartic in the state size. This renders RTRL training intractable for all but the smallest networks, even ones that are made highly sparse. We introduce the Sparse n-step Approximation (SnAp) to the RTRL influence matrix. SnAp only tracks the influence of a parameter on hidden units that are reached by the computation graph within $n$ timesteps of the recurrent core. SnAp with $n = 1$ is no more expensive than backpropagation but allows training on arbitrarily long sequences. We find that it substantially outperforms other RTRL approximations with comparable costs such as Unbiased Online Recurrent Optimization. For highly sparse networks, SnAp with $n = 2$ remains tractable and can outperform backpropagation through time in terms of learning speed when updates are done online.

## 1 Introduction

Recurrent neural networks (RNNs) have been successfully applied to a wide range of sequence learning tasks, including text-to-speech (Kalchbrenner et al., 2018), language modeling (Dai et al., 2019), automatic speech recognition (Amodei et al., 2016), translation (Chen et al., 2018) and reinforcement learning (Espeholt et al., 2018). RNNs have greatly benefited from advances in computational hardware, dataset sizes, and model architectures. However, the algorithm used to compute their gradients in almost all practical applications has not changed since the introduction of Back-Propagation Through Time (BPTT). The key limitation of BPTT is that the entire state history must be stored, meaning that the memory cost grows linearly with the sequence length. For sequences too long to fit in memory, as often occurs in domains such as language modelling or long reinforcement learning episodes, truncated BPTT (TBPTT) (Williams & Peng, 1990) can be used. Unfortunately the truncation length used by TBPTT also limits the duration over which temporal structure can be realiably learned.

Forward-mode differentiation, or Real-Time Recurrent Learning (RTRL) as it is called when applied to RNNs (Williams & Zipser, 1989), solves some of these problems. It doesn't require storage of any past network states, can theoretically learn dependencies of any length and can be used to update parameters at any desired frequency, including every step (i.e. fully online). However, its fixed storage requirements are $O(k \cdot |\theta|)$, where $k$ is the state size and $|\theta|$ is the number of parameters $\theta$ in the core. Perhaps even more daunting, the computation it requires is $O(k^2 \cdot |\theta|)$. This makes it impractical for even modestly sized networks. The advantages of RTRL have led to a search for more efficient approximations that retain its desirable properties, whilst reducing its computational and memory costs. One recent line of work introduces unbiased, but noisy approximations to the influence update. Unbiased Online Recurrent Optimization (UORO) (Tallec

& Ollivier, 2018) is an approximation with the same cost as TBPTT – $O(|\theta|)$ – however its gradient estimate is severely noisy (Cooijmans & Martens, 2019) and its performance has in practice proved worse than TBPTT (Mujika et al., 2018). Less noisy approximations with better accuracy on a variety of problems include both Kronecker Factored RTRL (KF-RTRL) (Mujika et al., 2018) and Optimal Kronecker-Sum Approximation (OK) (Benzing et al., 2019). However, both increase the computational costs to $O(k^3)$.

The last few years have also seen a resurgence of interest in sparse neural networks – both their properties (Frankle & Carbin, 2019) and new methods for training them (Evci et al., 2019). A number of works have noted their theoretical and practical efficiency gains over dense networks (Zhu & Gupta, 2018; Narang et al., 2017; Elsen et al., 2019). Of particular interest is the finding that scaling the state size of an RNN while keeping the number of parameters constant leads to increased performance (Kalchbrenner et al., 2018).

In this work we introduce a new sparse approximation to the RTRL influence matrix. The approximation is biased but not stochastic. Rather than tracking the full influence matrix, we propose to track only the influence of a parameter on neurons that are affected by it within $n$ steps of the RNN. The algorithm is strictly less biased but more expensive as $n$ increases. The cost of the algorithm is controlled by $n$ and the amount of sparsity in the Jacobian of the recurrent cell. We study the nature of this bias in Appendix C. Larger $n$ can be coupled with concomitantly higher sparsity to keep the cost fixed. This enables us to achieve the benefits of RTRL with a computational cost per step comparable in theory to BPTT. The approximation approaches full RTRL as $n$ increases. Our contributions are as follows:

- We propose SnAp – a practical approximation to RTRL, which is is applicable to both dense and sparse RNNs, and is based on the sparsification of the influence matrix.

- We show that parameter sparsity in RNNs reduces the costs of RTRL in general and SnAp in particular.

- We carry out experiments on both real-world and synthetic tasks, and demonstrate that the SnAp approximation: (1) works well for language modeling compared to the exact unapproximated gradient; (2) admits learning temporal dependencies on a synthetic copy task and (3) can learn faster than BPTT when run fully online.

## 2 BACKGROUND

We consider recurrent networks whose dynamics are governed by $h_t = f_\theta(h_{t-1}, x_t)$ where $h_t \in \mathbb{R}^k$ is the state, $x_t \in \mathbb{R}^a$ is an input, and $\theta \in \mathbb{R}^p$ are the network parameters. It is assumed that at each step $t \in \{1, ..., T\}$, the state is mapped to an output $y_t = g_\phi(h_t)$, and the network receives a loss $\mathcal{L}_t(y_t, y_t^*)$. The system optimizes the total loss $\mathcal{L} = \sum_t \mathcal{L}_t$ with respect to parameters $\theta$ by following the gradient $\nabla_\theta \mathcal{L}$. The standard way to compute this gradient is BPTT – running backpropagation on the computation graph "unrolled in time" over a number of steps $T$:

$$\nabla_\theta \mathcal{L} = \sum_{t=1}^T \frac{\partial \mathcal{L}}{\partial h_t} \frac{\partial h_t}{\partial \theta_t} = \sum_{t=1}^T \left( \frac{\partial \mathcal{L}}{\partial h_{t+1}} \frac{\partial h_{t+1}}{\partial h_t} + \frac{\partial \mathcal{L}_t}{\partial h_t} \right) \frac{\partial h_t}{\partial \theta_t} \tag{1}$$

The recursive expansion $\frac{\partial \mathcal{L}}{\partial h_t} = \frac{\partial \mathcal{L}}{\partial h_{t+1}} \frac{\partial h_{t+1}}{\partial h_t} + \frac{\partial \mathcal{L}_t}{\partial h_t}$ is the backpropagation rule. The slightly nonstandard notation $\theta_t$ refers to the copy of the parameters used at time $t$, but the weights are shared for all timesteps and the gradient adds over all copies.

### 2.1 REAL TIME RECURRENT LEARNING (RTRL)

Real Time Recurrent Learning (Williams & Zipser, 1989) computes the gradient as:

$$\nabla_\theta \mathcal{L} = \sum_{t=1}^T \frac{\partial \mathcal{L}_t}{\partial h_t} \frac{\partial h_t}{\partial \theta} = \sum_{t=1}^T \frac{\partial \mathcal{L}_t}{\partial h_t} \left( \frac{\partial h_t}{\partial \theta_t} + \frac{\partial h_t}{\partial h_{t-1}} \frac{\partial h_{t-1}}{\partial \theta} \right) \tag{2}$$

This can be viewed as an iterative algorithm, updating $\frac{\partial h_t}{\partial \theta}$ from the intermediate quantity $\frac{\partial h_{t-1}}{\partial \theta}$. To simplify equation 2 we introduce the following notation: have $J_t := \frac{\partial h_t}{\partial \theta}, I_t := \frac{\partial h_t}{\partial \theta_t}$ and $D_t := \frac{\partial h_t}{\partial h_{t-1}}$. $J$ stands for "Jacobian", $I$ for "immediate Jacobian", and $D$ for "dynamics". We sometimes refer to $J$ as the "influence matrix". The recursion can be rewritten $J_t = I_t + D_t J_{t-1}$.

**Cost analysis** $J_t$ is a matrix in $\mathbb{R}^{k \times |\theta|}$, which can be on the order of gigabytes for even modestly sized RNNs. Furthermore, performing the operation $D_t J_{t-1}$ involves multiplying a $k \times k$ matrix by a $k \times |\theta|$ matrix each timestep. That requires $|\theta|$ times more computation than the forward pass of the RNN core. To make explicit just how expensive RTRL is – this is a factor of roughly one million for a vanilla RNN with 1000 hidden units.

## 2.2 Truncated RTRL and stale Jacobians

In analogy to Truncated BPTT, one can consider performing a gradient update partway through a training sequence (at time $t$) but still passing forward a stale state *and* a stale influence Jacobian $J_t$ rather than resetting both to zero after the update. This enables more frequent weight updating at the cost of a staleness bias. The Jacobian $J_t$ becomes "stale" because it tracks the sensitivity of the state to old parameters. Experiments (section 5.2) show that this tradeoff can be favourable toward more frequent updates in terms of data efficiency. In fact, much of the RTRL literature assumes that the parameters are updated at every step $t$ ("fully online") and that the influence Jacobian is never reset, at least until the start of a new sequence. All truncated BPTT experiments in our paper pass forward a stale state if an update is done before the end of the sequence.

## 2.3 Sparsity in RNNs

One of the early explorations of sparsity in the parameters of RNNs (i.e. many entries of $\theta$ are exactly zero) was Ström (1997), where one-shot pruning based on weight magnitude with subsequent retraining was employed in a speech recognition task. The current standard approach to inducing sparsity in RNNs (Zhu & Gupta, 2018) remains similar, except that magnitude based pruning happens slowly over the course of training so that no retraining is required.

Kalchbrenner et al. (2018) discovered a powerful property of sparse RNNs in the course of investigating them for text-to-speech – for a constant parameter and flop budget sparser RNNs have *more* capacity per parameter than dense ones. This property has so far only been shown to hold when the sparsity pattern is adapted during training (in this case, with pruning). Note that parameter parity is achieved by simultaneously increasing the RNN state size and the degree of sparsity. This suggests that training large sparse RNNs could yield powerful sequence models, but the memory required to store the history of (now much larger) states required for BPTT becomes prohibitive for long sequences. In this paper, we use a fixed sparsity pattern rather than pruning (see Appendix B), for simplicity. In particular, we pick uniformly at random which indices of weight matrices to force to zero and hold this sparsity pattern constant over the course of training.

## 3 The Sparse n-Step Approximation (SnAp)

Our main contribution in this work is the development of an approximation to RTRL called the Sparse n-Step Approximation (SnAp) which reduces RTRL's computational requirements substantially.

SnAp imposes sparsity on $J$ even though it is in general dense. We choose the sparsity pattern to be the locations that are non-zero after $n$ steps of the RNN (Figure 1). We also choose to use the same pattern for all steps, though this is not a requirement. This means that the sparsity pattern of $J_t$ is known and can be used to reduce the amount of computation in the product $D_t J_{t-1}$. See Figure 2 for a visualization of the process. The costs of the resulting methods are compared in Table 1. We note an alternative strategy would be to perform the full multiplication of $D_t J_{t-1}$ and then only keep the top-k values. This would reduce the bias of the approximation but increase its cost.

More formally, we adopt the following approximation for all $t$:

$$(J_t)_{ij} \approx \begin{cases} (J_t)_{ij} & \text{if } (\theta_t)_j \text{ influences hidden unit } (h_{t+n})_i \\ 0 & \text{otherwise} \end{cases}$$

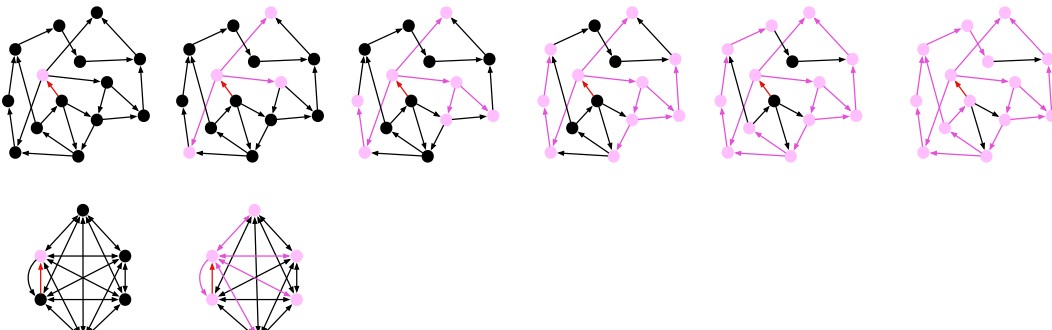

Figure 1: SnAp in dense (bottom) and sparse (top) graphs: As the figure proceeds to the right we propagate the influence of the red connection $i$ on the nodes $j$ of the graph through further RNN steps. Nodes are colored in pink if they are influenced on or before that step. The entry $J_{i,j}$ is kept if node $j$ is colored pink, but all other entries in row $i$ are set to zero. When $n = 1$ in both cases only one node is influenced. In the dense case the red connection influences all nodes when $n >= 2$.

## 3.1 SPARSE ONE-STEP APPROXIMATION (SNAP-1)

Even for a fully dense RNN, each parameter will in the usual case only immediately influence the single hidden unit it is directly connected to. This means that the immediate Jacobian $I_t$ tends to be extremely sparse. For example, a Vanilla RNN will have only one nonzero element per column, which is a sparsity level of $\frac{k-1}{k}$. Storing only the nonzero elements of $I_t$ already saves a significant amount of memory without making any approximations; $I_t$ is the same shape as the daunting $J_t$ matrix whereas the nonzero values are the same size as $\theta$.

$I_t$ can become more dense in architectures (such as GRU and LSTM) which involve the composition of parameterised layers within a single core step (see Appendix A for an in-depth discussion of the effect of gating architectures on Jacobian sparsity). In the Sparse One-Step Approximation, we only keep entries in $J_t$ if they are nonzero in $I_t$. After just two RNN steps, a given parameter has influenced every unit of the state through its intermediate influence on other units. Thus only SnAp with $n = 1$ is efficient for dense RNNs because $n > 1$ does not result in any sparsity in $J$; for dense networks SnAp-2 already reduces to full RTRL. (N.b.: SnAp-1 is also applicable to sparse networks.) Figure 1 depicts the sparse structure of the influence of a parameter for both sparse and fully dense cases.

SnAp-1 is effectively diagonal, in the sense that the effect of parameter $j$ on hidden unit $i$ is maintained throughout time, but ignoring the indirect effect of parameter $j$ on unit $i$ via paths through other units $i'$. More formally, it is useful to define $u(j)$ as the one component in the state $h_t$ connected directly to the parameter $j$ (which has at the other end of the connection some other entry $i'$ within $h_{t-1}$ or $x_t$). Let $i = u(j)$. The imposition of the one-step sparsity pattern means only the entry in row $i$ will be kept for column $j$ in $J_t$. Inspecting the update for this particular entry, we have

$$(J_t)_{ij} = (I_t)_{ij} + \sum_{m=1}^{n} (D_t)_{im}(J_{t-1})_{mj} = (I_t)_{ij} + (D_t)_{ii}(J_{t-1})_{ij} \tag{3}$$

The equality follows from the assumption that $(J_{t-1})_{mj} = 0$ if $m \neq i$. Diagonal entries in $D_t$ are thus crucial for this approximation to be expressive, such as those arising from skip connections.

## 3.2 OPTIMIZATIONS FOR FULL RTRL WITH SPARSE NETWORKS

When the RNN is sparse, the costs of even full (unapproximated) RTRL can be alleviated to a surprising extent; we save computation proportional to a factor of the sparsity *squared*. Assume a proportion $s$ of the entries in both $\theta$ and $D_t$ are equal to zero and refer to this number as "the level of sparsity in the RNN". For convenience, $d := 1 - s$. With a Vanilla RNN, this correspondence between parameter sparsity and dynamics sparsity holds exactly. For popular gating architectures such as GRU and LSTM the relationship is more complicated so we include empirical measurements of the computational cost in FLOPS (Table 2) in addition to the theoretical calculations here. More

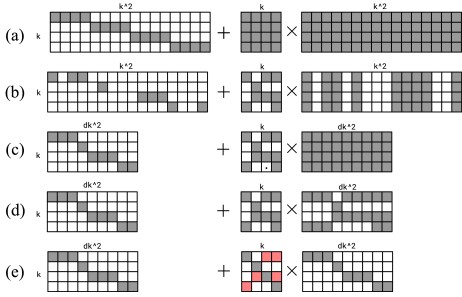

| Method | memory | time per step |
|---|---|---|
| BPTT | $Tk + p$ | $k^2 + p$ |
| UORO | $k + p$ | $k^2 + p$ |
| RTRL | $k + kp$ | $k^2 + k^2 p$ |
| Sparse BPTT | $Tk + dp$ | $d(k^2 + p)$ |
| Sparse RTRL | $k + dkp$ | $d(k^2 + dk^2 p)$ |
| SnAp-1 | $k + dp$ | $d(k^2 + p)$ |
| SnAp-2 | $k + d^2 kp$ | $d(k^2 + d^2 k^2 p)$ |

Figure 2: Depiction of RTRL, RTRL with sparsity and SnAp. White indicates zeros. (a) $I_t + D_t J_{t-1}$ (b) $I_t + D_t J_{t-1}$ when $W$ is sparse (c) $\tilde{I}_t + D_t \tilde{J}_{t-1}$ (d) SnAp-2 (e) SnAp-1. Rose colored squares are non-zero in $D_t$ but not used in updating $J_t$.

Table 1: Computational costs (up to a proportionality constant) of gradient calculation methods for dense and sparse RNNs. Below $T$ refers to the sequence length, $k$ the number of hidden units, $p$ the number of *dense* recurrent parameters, $s$ the level of sparsity, and $d = 1 - s$. The first term of the compute cost is for going forward and the second term is for either going backward or updating the influence matrix.

complex recurrent architectures involving attention (Rae et al., 2016) would require an independent mechanism for inducing sparsity in $D_t$; we leave this direction to future work and assume in the remainder of this derivation that sparsity in $\theta$ corresponds to sparsity in $D_t$.

If the sparsity level of $\theta$ is $s$, then so is the sparsity in $J$ because the columns corresponding to parameters which are clamped to zero have no effect on the gradient computation. We may extract the columns of $J$ containing nonzero parameters into a new dense matrix $\tilde{J}$ used in place of J everywhere with no effect on the gradient computation. We make the same optimization for $I_t$ and use the dense matrix $\tilde{I}_t$ in its place, leaving us with the update rule (depicted in Figure 2) :

$$\tilde{J}_t = \tilde{I}_t + D_t \tilde{J}_{t-1} \tag{4}$$

These optimizations taken together reduce the storage requirements by $\frac{1}{d}$ (because $\tilde{J}$ is $d$ times the size of $J$) and the computational requirements by $\frac{1}{d^2}$ because $D_t$ in the sparse matrix multiplication $D_t \tilde{J}_{t-1}$ has density $d$, saving us an extra factor of $\frac{1}{d}$.

### 3.3 Sparse $N$ Step Approximation (SnAp-$N$)

Even when $D_t$ is sparse, the computation "graph" linking nodes (neurons) in the hidden state over time should still be connected, meaning that $\tilde{J}$ eventually becomes fully dense because after enough iterations every (non-zero) parameter will have influenced every hidden unit in the state. Thus sparse approximations are still available in this setting and indeed required to obtain an efficient algorithm. For sparse RNNs, SnAp simply imposes additional sparsity on $\tilde{J}_t$ rather than $J_t$. SnAp-$N$ for $N > 1$ is both strictly less biased and strictly more expensive, but its costs can be reduced by increasing the degree $s$ of sparsity in the RNN. SnAp-2 is comparable with UORO and SnAp-1 if the sparsity of the RNN is increased so that $d < n^{-\frac{2}{3}}$, e.g. 99% or higher sparsity for a 1000-unit Vanilla RNN. If this level of sparsity is surprising, the reader is encouraged to see our experiments in Appendix B.

## 4 Related Work

SnAp-1 is actually similar to the original algorithm used to train LSTM (Hochreiter & Schmidhuber, 1997), which employed forward-mode differentiation to maintain the sensitivity to each parameter of a single cell unit, over all time. This exposition was expressed in terms coupled to the LSTM architecture whereas our formulation is general. SnAp-1 was also described in (Bellec et al., 2019) as eprop-1. The exposition in that paper goes into great depth regarding its biological plausibility and

relation to spiking neural networks and may be somewhat unfamiliar to readers from a pure machine learning background. The -1 postfix in eprop refers to it being the first of three present algorithms, not the number of connections as in SnAp. Biological plausibility of RTRL variants has also been studied in (Zenke & Neftci, 2021). An idea similar to SnAp was also proposed in (Bradbury, 1997), aiming to overcome poor local minima during optimization.

Random Feedback Local Online (Murray, 2019) (RFLO) amounts to accumulating $I_t$ terms in equation 4 whilst ignoring the product $D_t J_{t-1}$. It admits an efficient implementation through operating on $\tilde{I}_t$ as in section 3.2 but is strictly more biased than the approximations considered in this work and performs worse in our experiments. The original paper also used random matrices to propagate errors backward, thus avoiding the weight transport problem (Lillicrap et al., 2016). However, for a fair comparison, our re-implementation uses the same weights that are used for the forward pass, as in standard backpropagation. As mentioned in section 1, stochastic approximations to the influence matrix are an alternative to the methods developed in our work, but suffer from noise in the gradient estimator (Cooijmans & Martens, 2019). A fruitful line of research focuses on reducing this noise (Cooijmans & Martens, 2019), (Mujika et al., 2018), (Benzing et al., 2019).

It is possible to reduce the storage requirements of TBPTT using a technique known as "gradient checkpointing" or "rematerialization". This reduces the memory requirements of backpropagation by recomputing states rather than storing them. First introduced in Griewank & Walther (2000) and later applied specifically to RNNs in Gruslys et al. (2016), these methods are not compatible with the fully online setting where $T$ may be arbitrarily large as even the optimally small amount of re-computation can be prohibitive. For reasonably sized $T$, however, rematerialization is a straightforward and effective way to reduce the memory requirements of TBPTT, especially if the forward pass can be computed quickly.

## 5 EXPERIMENTS

We include experimental results on the real world language-modelling task WikiText103 (Merity et al., 2017) and the synthetic 'Copy' task (Graves et al., 2016) of simply repeating an observed binary string. Whilst the first is important for demonstrating that our methods can be used for real, practical problems, language modelling doesn't directly measure a model's ability to learn structure that spans long time horizons. The Copy task, however, allows us to parameterize exactly the temporal distance over which structure is present in the data. In terms of, respectively, task complexity and RNN state size (up to 1024) these investigations are considerably more "large-scale" than much of the RTRL literature.

### 5.1 WIKITEXT103

All of our WikiText103 experiments tokenize at the character (byte) level and use SGD to optimize the log-likelihood of the data. We use the Adam optimizer (Kingma & Ba, 2014) with default hyper-parameters $\beta_1 = 0.9$, $\beta_2 = 0.999$, and $\epsilon = 1e^{-8}$. We train on randomly cropped sequences of length 128 sampled uniformly with replacement and do not propagate state across the end-of-sequence boundary (i.e. no truncation). Results are reported on the standard validation set.

#### 5.1.1 LANGUAGE MODELLING WITH DENSE RNNs: SNAP-1

In this section, we refrain from performing a weight update until the end of a training sequence (see section 2.2) so that BPTT is the gold standard benchmark for performance, assuming the gradient is the optimal descent direction. The architecture is a Gated Recurrent Unit (GRU) network (Cho et al., 2014) with 128 recurrent units and a one-layer readout MLP mapping to 1024 hidden relu units before the final 256-unit softmax layer. The embedding matrix is not shared between the input and output. All weights are initialized from a truncated normal distribution with standard deviation equal to the inverse square root of the fan in. Learning curves in Figure 3 (Left) show that SnAp-1 outperforms RFLO and UORO, and that in this setting UORO fails to match the surprisingly strong baseline of not training the recurrent parameters at all and instead leaving them at their randomly initialized value. This random baseline is closely related to the Echo-State network (Jaeger, 2001), and the strong readout network is intended to help keep the comparison to this baseline fair.

#### 5.1.2 LANGUAGE MODELING WITH SPARSE RNNs: SNAP-1 AND SNAP-2

Here we use the same architecture as in section 5.1.1, except that we introduce 75% sparsity into the weights of the GRU, in particular the weight matrices (more sparsity levels are considered in

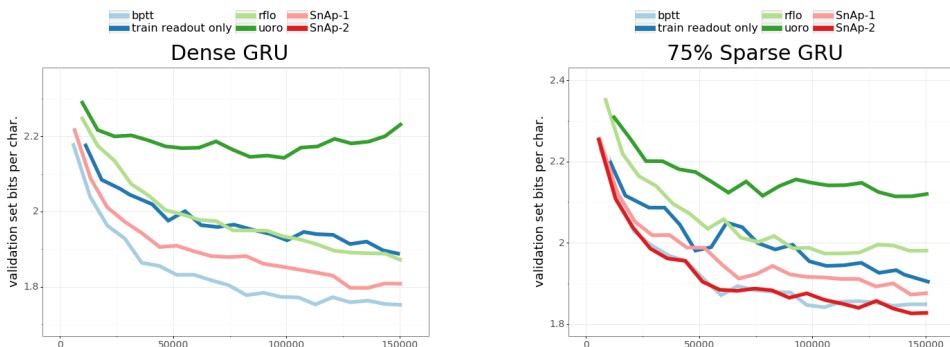

Figure 3: **Left:** Comparing various RTRL approximations based upon their ability to train a dense GRU network to do character-level language modelling. On the y-axis is Negative Log Likelihood. **Right:** Same as left with 75% parameter-sparsity.

later experiments). Biases are always kept fully dense. In order to induce sparsity, we generate a sparsity pattern uniformly at random and fix it throughout training. As would be expected because it is strictly less biased, Figure 3 (Right) shows that SnAp-2 outperforms SnAp-1 but only slightly. Furthermore, both closely match the (gold-standard) accuracy of a model trained with BPTT. Table 2 shows that SnAp-2 actually costs about 600x more FLOPs than BPTT/SnAp-1 at 75% sparsity, but higher sparsity substantially reduces FLOPs. It's unclear exactly how the cost compares to UORO, which though $O(|\theta|)$ does have constant factors required for e.g. random number generation, and additional overheads when approximations use rank higher than one.

## 5.2 Copy Task

Our experiments on the Copy task (Graves et al., 2016) aim to investigate the ability of the proposed sparse RTRL approximations to learn about temporal structure. In this synthetic task, a sequence of bits $b_t \in \{0, 1\}$ is presented one at a time, and then a special token is presented, denoting the end of the input pattern. Subsequently, the network receives a series of special tokens indicating that an output is desired, at which time it must output, one token at a time, the same binary string it received as input. Unlike language modelling, there is nothing going on in this problem except for (simple) temporal structure over a known temporal distance: the length of the input sequence.

We follow (Mujika et al., 2018) and adopt a curriculum-learning approach over the length $L$ of sequences to be copied, starting with $L = 1$. When the average bits per character of a training minibatch drops below 0.15, we increment $L$ by one. We sample the length of target sequences uniformly between $[max(L - 5, 1), L]$ as in previous work. We measure performance versus 'data-time', i.e. we give each algorithm a time budget in units of the cumulative number of tokens seen throughout training. A consequence of this scheme is that full BPTT is no longer an upper bound on performance because, for example, updating once on a sequence of length 10 with the true gradient may yield slower learning than updating twice on two consecutive sequences of length 5, with truncation.

In these experiments we examine SnAp performance for multiple sparsity levels and recurrent architectures including Vanilla RNNs, GRU, and LSTM. Table 2 includes the architectural details. The sparsity pattern is again chosen uniformly at random. As a result, comparison between sparsity levels is discouraged. For each configuration we sweep over learning rates in $\{10^{-2.5}, 10^{-3}, 10^{-3.5}, 10^{-4}\}$ and compare average performance over three seeds with the best chosen learning rate (all methods performed best with learning rate $10^{-3}$). The minibatch size was 16. We train with either full unrolls or truncation with $T = 1$. This means that the RTRL approximations update the network weights at every timestep and persist the RNN state along with a stale Jacobian (see section 2.2).

**Fully online training** One striking observation is that Truncated BPTT completely fails to learn temporal structure in the fully online ($T = 1$) regime. Interestingly, the SnAp methods perform *better* with more frequent updates. Compare solid versus dotted lines of the same color in Figure 4. Fully online SnAp-2 and SnAp-3 mostly outperform or match BPTT for training LSTM and GRU architectures despite the "staleness" noted in Section 2.2. We attribute this to the hypothesis

advanced in the RTRL literature that Jacobian staleness can be mitigated with small learning rates but leave a more thorough investigation of this phenomenon to future work.

**Bias versus computational expense** For SnAp there is a tradeoff between the biasedness of the approximation and the computational costs of the algorithm. We see that correspondingly, SnAp-1 is outperformed by SnAp-2, which is in turn outperformed by SnAp-3 in the Copy experiments. The RFLO baseline is even more biased than SnAp-1, but both methods have comparable costs. SnAp-1 significantly outperforms RFLO in all of our experiments. The nature of the bias introduced by SnAp is investigated in Appendix C.

**Empirical FLOPs requirements** Here we augment the asymptotic cost calculations from Table 1 with empirical measurements of the FLOPs, broken out by architecture and sparsity level in Table 2. Gating architectures require a high degree of parameter sparsity in order to keep a commensurate amount of of Jacobian sparsity due to the increase in density brought about by composing linear maps with different sparsity patterns (see Appendix A). For instance, the 75% sparse GRU considered in the experiments from Section 5.1.2 lead to SnAp-2 parameter Jacobian that is only 70.88% sparse. With SnAp-3 it becomes much less sparse – only 50%. This may partly explain why SnAp performs best compared to BPTT in the LSTM case (Figure 4), though it still significantly outperforms BPTT in the high sparsity regime when SnAp-2 becomes practical. Also, LSTM is twice as costly to train with RTRL-like algorithms because it has two components to its state, requiring the maintenance of twice as many jacobians and the performance of twice as many jacobian multiplications (Equations 3/5). For a 75% sparse LSTM, the SnAp-2 Jacobian is much denser at 38.5% sparsity and SnAp-3 has essentially reached full density (so it is as costly as RTRL).

Figure 4 also shows that for Vanilla RNNs, increasing $n$ improves performance, but SnAp does not outperform BPTT with this architecture. In summary, Increasing $n$ improves performance but costs more FLOPs.

| Architecture | ‖ | Vanilla | | ‖ | GRU | | ‖ | LSTM | |
|---|---|---|---|---|---|---|---|---|---|---|
| Number of Units | 128 | 256 | 512 | 128 | 256 | 512 | 128 | 256 | 512 |
| Param. Sparsity | 75.0% | 93.8% | 98.4% | 75% | 93.8% | 98.4% | 75.0% | 93.8% | 98.4% |
| SnAp-2 J Sparsity | 83.0% | 95.6% | 98.9% | 70.9% | 91.1% | 97.8% | 38.5% | 79.9% | 95.1% |
| SnAp-3 J Sparsity | 33.3% | 59.2% | 92.8% | 50.0% | 52.5% | 71.6% | 2.4% | 5.9% | 38.7% |
| SnAp-1 vs BPTT | 1x | 1x | 1x | 1x | 1x | 1x | 2x | 2x | 2x |
| SnAp-2 vs BPTT | 349x | 90.4x | 22.1x | 597x | 183x | 44.8x | 2518x | 824.8x | 200.1x |
| SnAp-3 vs BPTT | 1365x | 835.8x | 147.5x | 1024x | 972x | 582x | 3996x | 3855x | 2513x |
| SnAp-2 vs RTRL | 0.17x | 0.044x | 0.011x | 0.291x | 0.089x | 0.022x | 0.615x | 0.201x | 0.049x |

Table 2: Empirical computational costs of SnAp, determined by the sparsity level in the Jacobians. The "X vs BPTT" rows express the FLOPS requirements of X as a multiple of BPTT training FLOPs. The "SnAp-2 vs RTRL" row shows the FLOPS requirements of SnAp-2 as a multiple of those required by optimized Sparse RTRL (section 3.2). Lower is better for all of these entries.

## 6 CONCLUSION

We have shown how sparse operations can make a form of RTRL efficient, especially when replacing dense parameter Jacobians with approximate sparse ones. We introduced SnAp-1, an efficient RTRL approximation which outperforms comparably-expensive alternatives on a popular language-modeling benchmark. We also developed higher orders of SnAp including SnAp-2 and SnAp-3, approximations tailor-made for sparse RNNs which can be efficient in the regime of high parameter sparsity, and showed that they can learn temporal structure considerably faster than even full BPTT.

Our results suggest that training very large, sparse RNNs could be a promising path toward more powerful sequence models trained on arbitrarily long sequences. This may prove useful for modelling whole documents such as articles or even books, or reinforcement learning agents which learn over an entire lifetime rather than the brief episodes which are common today.

A few obstacles stand in the way of scaling up our methods further:

- The need for a high-performing sparse training strategy that does not require dense gradient information.

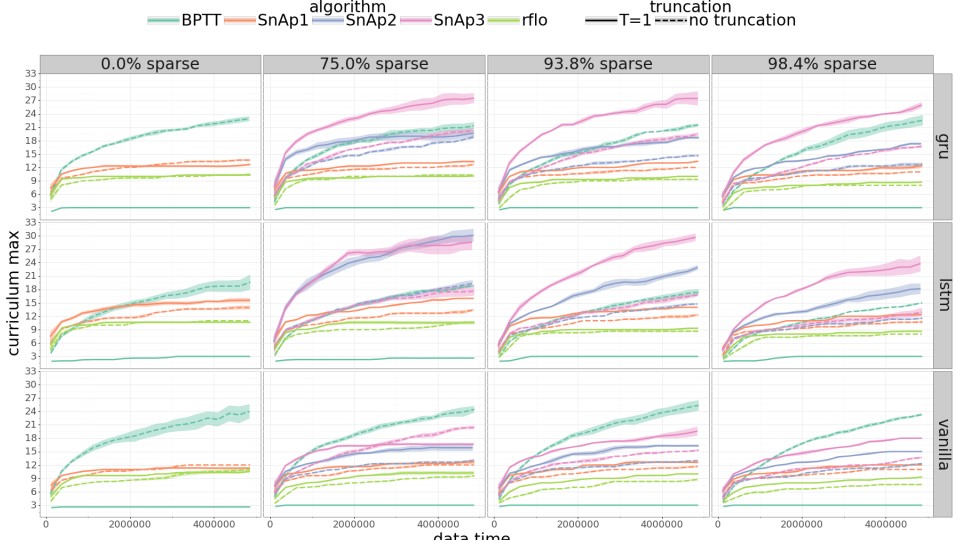

Figure 4: Copy task results by sparsity and architecture. Solid lines indicate that updates are done fully online (at every step). Dotted lines indicate that updates are done at the end of a sequence. See the heading "Fully online training" within section 5.2 for more details.

- Sparsity support in both software and hardware that enables better realization of the theoretical efficiency gains of sparse operations.

It may also be fruitful to further develop our methods for hybrid models combining recurrence and attention (Dai et al., 2019; Rae et al., 2016) or even feedforward architectures with tied weights (Lan et al., 2019) (Dehghani et al., 2018).

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

APPENDIX A: JACOBIAN SPARSITY OF GRUS AND LSTMS

Unlike vanilla RNNs whose dynamics Jacobian $D_t$ has sparsity exactly equal to the sparsity of the weight matrix, GRUs and LSTMs have inter-cell interactions which increase the Jacobians' density. In particular, the choice of GRU variant can have a very large impact on the increase in density. This is relevant to the "dynamics" jacobian $D_t$ and the parameter jacobians $I_t$ and $J_t$.

Consider a standard formulation of LSTM.

$$
\begin{aligned}
i_t &= \sigma(W_{ii}x_t + W_{hi}h_{t-1} + b_i) \\
f_t &= \sigma(W_{if}x_t + W_{hf}h_{t-1} + b_f) \\
o_t &= \sigma(W_{io}x_t + W_{ho}h_{t-1} + b_o) \\
g_t &= \phi(W_{ig}x_t + W_{hg}h_{t-1} + b_g) \\
c_t &= f_t \odot c_{t-1} + i_t \odot g_t \\
h_t &= o_t \odot \phi(c_t)
\end{aligned}
\tag{5}
$$

Looking at LSTM's update equations, we can see that an individual parameter $(W, b)$ will only directly affect one entry in each gate ($i_t$, $f_t$, $o_t$) and the candidate cell $g_t$. These in turn produce the next cell $c_t$ and next hidden state $h_t$ with element-wise operations ($\sigma$ is the sigmoid function applied element-wise and $\phi$ is usually hyperbolic tangent). In this case Figure 1 is an accurate depiction of the propagation of influence of a parameter as the RNN is stepped.

However, for a GRU there are multiple variants in which a parameter or hidden unit can influence many more units of the next state. The original variant (Cho et al., 2014) is as follows:

$$
\begin{aligned}
z_t &= \sigma(W_{iz}x_t + W_{hz}h_{t-1} + b_z) \\
r_t &= \sigma(W_{ir}x_t + W_{hr}h_{t-1} + b_r) \\
a_t &= \phi(W_{ia}x_t + W_{ha}(r_t \odot h_{t-1}) + b_a) \\
h_t &= (1 - z_t) \odot h_{t-1} + z_t \odot a_t
\end{aligned}
\tag{6}
$$

For our purposes the main thing to note is that the parameters influencing $r_t$ further influence every unit of $a_t$ because of the matrix multiplication by $W_{ha}$. They therefore influence every unit of $h_t$ within one recurrent step, which means that the dynamics jacobian $D_t$ is fully dense and the immediate parameter jacobian $I_t$ for $W_{ir}$, $W_{hr}$, and $b_r$ are all fully dense as well.

An alternative formulation which was popularized by Engel, and also used in the CuDNN library from NVIDIA is given by:

$$
\begin{aligned}
z_t &= \sigma(W_{iz}x_t + W_{hz}h_{t-1} + b_z) \\
r_t &= \sigma(W_{ir}x_t + W_{hr}h_{t-1} + b_r) \\
a_t &= \phi(W_{ia}x_t + r_t \odot W_{ha}h_{t-1} + b_a) \\
h_t &= (1 - z_t) \odot h_{t-1} + z_t \odot a_t
\end{aligned}
\tag{7}
$$

The second variant has moved the reset gate after the matrix multiplication, thus avoiding the composition of parameterized linear maps within a single RNN step. As the modeling performance of the two variants has been shown to be largely the same, but the second variant is faster and results in sparser $D_t$ and $I_t$, we adopt the second variant throughout this paper.

APPENDIX B: SPARSITY STRATEGY

Our experiments do not use state-of-the-art strategies for inducing sparsity because there is no such strategy compatible with SnAp at the time of writing. The requirement of a dense gradient in Evci et al. (2019) and Zhu & Gupta (2018) prevents the use of the optimization in Equation 4, which is strictly necessary to fit the RTRL training computations on accelerators without running out of memory.

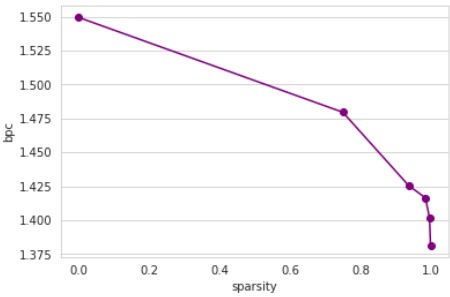

| units | bpc | $\theta$ sparsity | $|\theta|$ |
|---|---|---|---|
| base | 1.55 | 0% | 1x |
| 2x | 1.48 | 75% | 1x |
| 4x | 1.43 | 93.75% | 1x |
| 8x | 1.42 | 98.4% | 1x |
| 16x | 1.40 | 99.6% | 1x |
| **32x** | **1.38** | **99.9%** | **1x** |
| 2.5x | 1.39 | 0% | 6.25x |

Figure 5: BPC vs Sparsity for Constant Parameter Count. Shows the same results as Table 3. The biggest, sparsest GRU performs better than a dense network with 6.25x as many (nonzero) parameters.

Table 3: Final performance of sparse WikiText103 language modeling GRU networks trained with progressive pruning. Each row represents a single training run. The 'bpc' column gives the validation set negative log-likelihood in units of bits per character. The $|\theta|$ column gives the number of parameters in the network as a multiple of the 'base' 128-unit model.

To further motivate the development of sparse training strategies that do not require dense gradients, we show that larger sparser networks trained with BPTT and magnitude pruning monotonically outperform their denser counterparts in language modelling, when holding the number of parameters constant. This provides more evidence for the scaling law observed in Kalchbrenner et al. (2018).

The experimental setup is identical to the previous section except that all networks are trained with full BPTT. To hold the number of parameters constant, we start with a fully dense 128-unit GRU. We make the weight matrices 75% sparse when the network has 256 units, 93.8% sparse when the network has 512 units, 98.4% when the network has 1024 units, and so on. The sparsest network considered has 4096 units and over 99.9% sparsity, and performed the best. Indeed it performed better than a dense network with 6.25x as many parameters (Figure 5). Pruning decisions are made on the basis of absolute value every 1000 steps, and the final sparsity is reached after 350,000 training steps.

APPENDIX C: ANALYSIS OF THE BIAS INTRODUCED BY SNAP

Finally, we examine the empirical magnitudes of entries which are nonzero in the true, unapproximated influence matrix but set to zero by SnAp. For the benefit of visualization we train a small GRU network (8 units, 75% sparsity) on a non-curriculum variant of the Copy-task with target sequences fixed in length to 16 timesteps. This enables us to measure and display the bias of SnAp. The influence matrix considered is the final value after processing an entire sequence. The network is optimized with full (untruncated) BPTT. We find (Table 4) that at the beginning of training the influence entries ignored by SnAp are small in magnitude compared to those kept, even after the influence has had many RNN iterations to fill in.

This analysis complements the experimental results concerning how useful the approximate gradients are for learning; instead it shows where — and by how much — the sparse approximation to the influence differs from the true accumulated influence. Interestingly, despite the strong task performance of SnAp, the magnitude of ignored entries in the influence matrix is not always small (see Figure 6). The accuracy, as measured by such magnitudes, trends downward over the course of training. We speculate that designing methods to temper the increased bias arising later in training may be beneficial but leave this to future work.

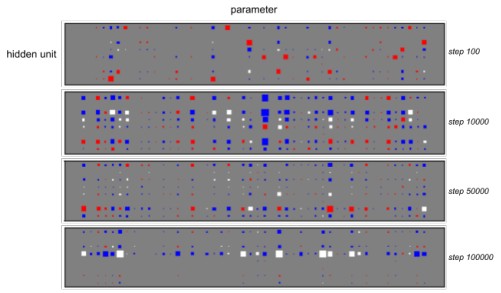

| Training Step | SnAp-1 | SnAp-2 |
|---|---|---|
| 100 | 1.0E-2 (73%) | 4.0E-3 (97%) |
| 5k | 2.3E-1 (22%) | 2.6E-1 (78%) |
| 10k | 1.1E-1 (23%) | 1.2E-1 (85%) |
| 50k | 3.3E-1 (34%) | 2.5E-1 (87%) |
| 100k | 2.4E-1 (6%) | 6.5E-1 (51%) |

Figure 6: Influence matrix for 75% sparse GRU with 8 units after processing a full sequence with 35 timesteps (target length 16), at various points during training ("step" corresponds to training step, not e.g. the step within a sequence). This Hinton-diagram shows the magnitude of an entry with the size of a square. Grey entries are near zero. Entries filled in with red are those included by SnAp-1. Blue entries are those included by SnAp-2, and white ones are ignored by both approximations.

Table 4: Approximation Quality of SnAp-1 and SnAp-2. Average magnitudes in the influence matrix versus whether or not they are kept by an approximate method. The "SnAp-1" and "SnAp-2" columns show the average magnitude of entries kept by the SnAp-1 and SnAp-2 approximations respectively. In parentheses is the sum of the magnitudes of entries in this category divided by the sum of all entry magnitudes in the influence matrix.

APPENDIX D: CODE SNIPPET FOR SNAP-1

We include below a code snippet showing how RTRL and SnAp can be implemented in Jax (Bradbury et al., 2018). While it is real and working Jax code, this is just a sketch for pedagogical purposes and does not take full advantage of the optimizations in section 3.2.

Please take note of the license at the top of the snippet.

```
# Copyright The Authors of "Practical Real Time Recurrent Learning
# with a Sparse Approximation to the Jacobian", 2020
# SPDX-License-Identifier: Apache-2.0
import jax
import jax.numpy as jnp

def get_fwd_and_update_influence_func(core_f, use_snap1_approx=False):
  """Transform core_f into a one which maintains influence jacobian w/ RTRL."""

  def fwd_and_update_influence(prev_infl, params, state, inpt):
    # Run the forward pass on a batch of data.
    batched_model_fn = jax.vmap(lambda s, i: core_f(params, s, i))
    f_out, state_new = batched_model_fn(state, inpt)

    # Compute jacobians of state w.r.t. prev state and params.
    jac_fn = jax.jacrev(lambda p, s, i: core_f(p, s, i)[1], argnums=(0, 1))
    batched_jac_fn = jax.vmap(lambda s, i: jac_fn(params, s, i))
    p_jac, s_jac = batched_jac_fn(state, inpt)

    # Update the influence matrix according to RTRL learning rule.
    new_infl = jax.tree_multimap(
        lambda j_i, infl_i: j_i + jnp.einsum('bHh,bh...->bH...', s_jac, infl_i),
        p_jac, prev_infl)

    # SnAp-1: Keep only the entries of the influence matrix which are nonzero
    # after a single core step. This is not an efficient implementation.
    if use_snap1_approx:
      onestep_infl_mask = jax.tree_map(
          lambda t: (jnp.abs(t) > 0.).astype(jnp.float32), p_jac)
```

```python
        new_infl = jax.tree_multimap(
            lambda matrix, mask: matrix * mask, new_infl, onestep_infl_mask)

      return f_out, state_new, new_infl
    return fwd_and_update_influence

def compute_gradients(influence_nest, delta):
  grads = jax.tree_map(
      lambda influence_i: jnp.einsum('bH...,bH->...', influence_i, delta),
      influence_nest)
  return grads

def make_zero_infl(param_exemplar, state_exemplar):
  def make_infl_for_one_state(t):
    return jax.tree_map(
        lambda p: jnp.zeros(shape=list(t.shape) + list(p.shape)),
        param_exemplar)
  infl = jax.tree_map(make_infl_for_one_state, state_exemplar)
  return infl

def get_rtrl_grad_func(core_f, readout_f, loss_f, use_snap1_approx):
  """Transform functions into one which computes the gradient via RTRL."""
  fwd_and_update_influence = get_fwd_and_update_influence_func(
      core_f, use_snap1_approx=use_snap1_approx)

  def rtrl_grad_func(core_params, readout_params, state, data):
    def rtrl_scan_func(carry, x):
      """Function which can be unrolled with jax.lax.scan."""
      # Unpack state and input.
      old_state, infl_acc, core_grad_acc, readout_grad_acc, loss_acc = carry
      inpt, targt, msk = x

      # Update influence matrix.
      h_t, new_state, new_infl_acc = fwd_and_update_influence(
          infl_acc, core_params, old_state, inpt)

      # Compute output, loss, and backprop gradients for RNN state.
      def step_loss(ps, h, t, m):
        """Compute the loss for one RNN step."""
        y = readout_f(ps, h)
        return loss_f(y, t, m), y
      step_out_and_grad_func = jax.value_and_grad(
          step_loss, has_aux=True, argnums=(0, 1))
      step_out, step_grad = step_out_and_grad_func(
          readout_params, h_t, targt, msk)
      loss_t, y_out = step_out
      readout_grad_t, delta_t = step_grad

      # Update accumulated gradients.
      core_grad_t = compute_gradients(new_infl_acc, delta_t)
      new_core_grad_acc = jax.tree_multimap(
          jnp.add, core_grad_acc, core_grad_t)
      new_readout_grad_acc = jax.tree_multimap(
          jnp.add, readout_grad_acc, readout_grad_t)

      # Repack carried state and return output.
      new_carry = (new_state, new_infl_acc,
                   new_core_grad_acc, new_readout_grad_acc, loss_acc + loss_t)
      return new_carry, y_out

    zero_infl = make_zero_infl(core_params, state)
    zero_core_grad = jax.tree_map(jnp.zeros_like, core_params)
```

```
    zero_readout_grad = jax.tree_map(jnp.zeros_like, readout_params)
    final_carry, output_seq = jax.lax.scan(
        rtrl_scan_func,
        init=(state, zero_infl, zero_core_grad, zero_readout_grad, 0.0),
        xs=(data['input_seq'], data['target_seq'], data['mask_seq']))
    final_state, _, core_grads, readout_grads, loss = final_carry
    return (loss, (final_state, output_seq)), (core_grads, readout_grads)
return rtrl_grad_func
```

