# OpenReview forum: "Practical Real Time Recurrent Learning with a Sparse Approximation"
_ICLR.cc/2021/Conference — ICLR 2021 Spotlight_

### Official Review · AnonReviewer4 · 2020-10-28

**Rating:** 6
**Confidence:** 4

**Review:**

This work presents a method, named SnAp, that takes Real-time Recurrent Learning derivations and proposes to approximate its computations with sparse approximations to make them more computational tractable. The method is an alternative to overcome the truncation in backpropagation through time (BPTT) over long term temporal structure. The method assumes a sparsity pattern on the parameters of the network that leads to the relationship on how the gradients could be updated. Finally, the method is evaluated against BPTT, UORO and RTRL on character-level language modeling and a copy task.

=================

The method is simple and a complexity analysis has been included. The experimental section seems limited in only showing a performance comparison with other methods. A better analysis in aspects of the method (like capacity for learning long-term temporal patterns) is lacking.

==================

When does the Snap/RTRL is not applicable? Are there cases where BPTT is applicable and Snap/RTRL is not? It would be nice to clarify such cases so readers can understand when Snap, RTRL, or BPTT are the right or better solutions.

The motivation behind the method is to overcome the limitation of the truncation behind BPTT for learning long term temporal structure. However this seems to be evaluated with very short sequences overall (128 for language modeling and maybe less than that for the copy task, the length is not present). How well the Snap method would work when training for sequences of thousands of elements, where BPTT is well known to struggle?

The relationship between neurons can vary every time the parameters are updated. Is it assumed to be fixed over the entire training? What would happen if the pattern is updated every few gradient updates? Also, complexities for Snap in table 1 don’t seem to consider the computational time of computing the sparsity pattern (even for a random pattern).

When n is big, the experimental results show a better and competitive performance to BPTT. However, in such cases (like Snap-3) the computational cost becomes very expensive compared to BPTT by at least 2 orders of magnitude, and the matrices become more dense. What are the results of TBPTT when stateful training is applied and how do they compare in such case? Also, the copy task has been solved with fewer neurons in previous works.

The Copy task details are unclear for someone that doesn’t know the work from Mujika et al., 2018. Please describe all the details. What is the length L used in the experiments? What is the length of the overall sequence? What does “data time” mean in the plots of Figure 4?

==================

My concerns behind the limited practicality of the method, and the limited experimental results given the hypothesis that the method can learn long-term temporal patterns. These are my considerations not to accept this paper.

==================

Minor issues:

-Use bigger fonts in the plots, and diagrams.

-In 5.1, do you use SGD or Adam?

-Figure 3: leave space between caption and figures

-Table 1 caption: “Below” -> “Above”

-It would be nice to mention the relationship between |\theta| and k, for each recurrent cell case.

---

> ### Author Response · Authors · 2020-11-17
> **Official Reply**
>
> We thank reviewer #4 for their comments.
>
> Reviewer #4 seeks clarity on when we should use SnAp instead of BPTT/RTRL in practice. This paper isn’t about a new method that should be used in production immediately, but the same is true of much (all?) of the literature on making Real Time Recurrent Learning efficient. And yet ICLR has served as one of the leading venues for this line of research in recent years [see e.g.: https://openreview.net/forum?id=rJQDjk-0b, https://openreview.net/forum?id=ryGfnoC5KQ].
>
> Indeed, as the self-identified expert reviewers for our paper have commented, this is basic research into an important and active research topic in the Deep Learning community at large, (as well as in the neuroscience community): the development of temporal learning algorithms that are scalable to long sequences, large networks, and fully online learning. We agree with these reviewers (#1 and #2) that our paper has strong merit as a contribution to this fruitful line of research.
>
> The most important criticism from Reviewer #4 is that we haven’t demonstrated the capacity for learning long-term temporal patterns. We respectfully disagree: the Copy task experiments show exactly the length of time over which SnAp can capture temporal structure when training a wide array of reasonably-sized recurrent networks (Vanilla, GRU, and LSTM).
>
> We agree that it would be fruitful for future work to extend these investigations, as well as the Language Modelling ones to longer sequences. In our view this is a matter of scaling up the method to larger networks, because the ability of a recurrent network to capture long term temporal structure can be bottlenecked by the architecture size in terms of hidden units and the number of parameters, as well as the learning algorithm.
>
> In the paper’s Conclusion, we have clearly set out the software and hardware barriers to scaling up our methods further and believe that our experiments are a respectable first step. We can push further once the aforementioned engineering challenges are ameliorated.
>
> Below we resolve smaller matters which this reviewer wanted clarified, one-by-one.
>
> > The relationship between neurons can vary every time the parameters are updated. Is it assumed to be fixed over the entire training?... Also, complexities for Snap in table 1 don’t seem to consider the computational time of computing the sparsity pattern (even for a random pattern).
>
> In our experiments, the sparsity pattern is chosen uniformly and then fixed throughout training, as explained in the final sentence of section 2.3. This strategy is simple and compatible with our optimizations to RTRL (section 3.2). The costs of picking a random sparsity pattern end up being negligible compared to the training run just like other computation done for network initialization. There is an active literature investigating methods for adapting the sparsity pattern and it is far from a solved problem.  Combining dynamic sparsity patterns [SET, RigL, etc.] with our work would be an interesting future direction.
>
> > The Copy task details are unclear…
>
> We agree that the description has become quite terse due to space limits but we have added more detail in the latest revision: thanks for the suggestion! For a more thorough description of the task, please see [1, section 4.2].
>
> > Minor issues…
>
> Thanks also for catching these, we will update the manuscript.
>
> [1] Graves, A., Wayne, G. & Danihelka, I. Neural Turing machines. Preprint at http://arxiv.org/abs/1410.5401 (2014).

---

> > ### Comment · AnonReviewer4 · 2020-11-23
> > **Follow up questions**
> >
> > ### Follow up Questions:
> > Would like to thank the authors for taking time to answer my previous questions.
> >
> > First, I would like to clarify that I agree with other reviewers and the authors that *the ideas* in this work are a current research topic in this community and do have merit. Please, keep in mind that the whole content of your work is reviewed, not only ideas.
> >
> > > Reviewer #4 seeks clarity on when we should use SnAp instead of BPTT/RTRL in practice. This paper isn’t about a new method that should be used in production immediately, but the same is true of much (all?) of the literature on making Real Time Recurrent Learning efficient.
> >
> > My suggestion was to specify **in the text** what falls under the problems that can be solved with SnAp. If the same problems as RTRL can be solved, then a simple sentence or two stating it should suffice.
> >
> > > The most important criticism from Reviewer #4 is that we haven’t demonstrated the capacity for learning long-term temporal patterns. We respectfully disagree: the Copy task experiments show exactly the length of time over which SnAp can capture temporal structure when training a wide array of reasonably-sized recurrent networks (Vanilla, GRU, and LSTM).
> >
> > SnAp seems to be learning temporal patterns, however, can you explain why are these **long-term** dependencies?
> > Section 5.1 trains with sequences in language modeling of $128$ characters long. The copy task in section 5.2, seems to contain much less elements for the longest sequences. Assuming Mujika et al. setup with $2L+3$ for a sequence length, with max length of $\sim 63$ elements. For instance, [2] discuss a copy task for sequences with 1000 elements with an LSTM.
> >
> > > We agree that the description has become quite terse due to space limits but we have added more detail in the latest revision: thanks for the suggestion! For a more thorough description of the task, please see [1, section 4.2].
> >
> > Are the authors implementing the "repeat copy task" [1, section 4.2] or the "copy task" [1, section 4.1]? The title of section 5.2 reads "Copy Task" and there is no mentioning about the repetitions in the text.
> >
> > * What is the y-axis showing in Figure 4? Is the maximum $L$ achieved?
> > * Is stateful or stateless training used for TBPTT with $T = 1$ in the copy task? And in the character-level language modeling task?
> >
> > I will be happy to modify my score, if the questions above are answered and the paper improved accordingly.
> > If submitted, happy to look at code for clarifications instead.
> >
> > ### References:
> >
> > [1] Graves, A., Wayne, G. & Danihelka, I. Neural Turing machines. Preprint at http://arxiv.org/abs/1410.5401 (2014)
> >
> > [2] Tallec C. and Ollivier Y. Can Recurrent Neural Networks warp time?. ICLR 2018.

---

> > > ### Author Response · Authors · 2020-11-24
> > > **Official reply to missed point in first review.**
> > >
> > > Thanks very much to reviewer #4 for taking the time to read our reply thoroughly, double-check the references, and give us a thoughtful response. We have addressed your comments in a new revision and uploaded it to OpenReview.
> > >
> > > First, we noticed that we missed a point in your initial review.
> > >
> > > > When n is big, the experimental results show a better and competitive performance to BPTT. However, in such cases (like Snap-3) the computational cost becomes very expensive compared to BPTT by at least 2 orders of magnitude, and the matrices become more dense.
> > >
> > > We agree that the case for using SnAp-N in practice becomes diminished as N grows large (e.g. 3 or greater), because the costs become comparable to full RTRL and therefore much more costly than BPTT except in the regime of extremely high sparsity.
> > >
> > > For SnAp-2, the asymptotic numbers in Table 1 show that in theory the computation costs can be reduced to the level of BPTT (or lower) by making the network sparse enough. Quoting section 3.3, the costs become comparable to BPTT if “if the sparsity of the RNN is increased so that $d < n^{\frac{-2}{3}}$ , e.g. 99% or higher sparsity for a 1000-unit Vanilla RNN.” Snap-2 does do quite well in e.g. language modelling (Figure 2) and Copy (Figure 4), outperforming BPTT for training sparse LSTMs in terms of learning speed.
> > >
> > > Appendix B was included to motivate the sparsity level/network size we’d need to fully realize a performance win in practice, but we haven’t yet managed to scale SnAp to this regime.

---

> > > > ### Author Response · Authors · 2020-11-24
> > > > **Official reply to follow-up questions.**
> > > >
> > > > Follow-up questions addressed point-by-point.
> > > >
> > > > > What are the results of TBPTT when stateful training is applied and how do they compare in such case?
> > > >
> > > > Apologies for the lack of clarity here, whenever we truncate we do “stateful training”, i.e. we pass the RNN state forward even though we have done a weight update in the interim. We have updated section 2.2 of our paper to be explicit that all “truncated BPTT” experiments in our paper are “stateful” in the sense  that they pass forward the RNN state across truncation boundaries.
> > > >
> > > > > Also, the copy task has been solved with fewer neurons in previous works
> > > >
> > > > The big difference in our work is that these networks are highly sparse, so the number of neurons is not an apples-to-apples comparison of network capacity.  This also depends on the version of the copy task in question.  It seems there is some confusion -- please see our response below as to why the tasks in the “Can recurrent networks warp time” paper are not equivalent to the copy task used here.
> > > >
> > > > > My suggestion was to specify in the text what falls under the problems that can be solved with SnAp. If the same problems as RTRL can be solved, then a simple sentence or two stating it should suffice.
> > > >
> > > > Thanks, we agree this could be made more explicit and have added a sentence to this effect in the introduction.
> > > >
> > > > > Are the authors implementing the "repeat copy task" [1, section 4.2] or the "copy task" [1, section 4.1]? The title of section 5.2 reads "Copy Task" and there is no mentioning about the repetitions in the text.
> > > >
> > > > Good catch, it is indeed “Copy” not “Repeat Copy” so we should indeed have referred you to section 4.1, not section 4.2 in the previous reply. As in Mujika et al [1], the sequence is only copied once, there are no repetitions. Sequences are indeed length 2L + 3.
> > > >
> > > > > What is the y-axis showing in Figure 4? Is the maximum  L achieved?
> > > >
> > > > Yes exactly, an (x, y) point in that plot shows the level of L=y achieved after data_time=x training tokens have been processed.
> > > >
> > > > > Is stateful or stateless training used for TBPTT with T=1 in the copy task? And in the character-level language modeling task?
> > > >
> > > > Stateful a.k.a “truncated” BPTT. As described in section 2.2 (updated). In the language modelling experiments we always do full backpropagation through time on the whole sequence, which is why BPTT is a “gold standard” upper limit on performance.
> > > >
> > > > > For instance, [2] discuss a copy task for sequences with 1000 elements with an LSTM.
> > > >
> > > > The copy task in that paper is much different from the one in our paper and the ones we have been discussing in [1, 2]. Rather than copying length 1000 sequences, that work is copying length-10 sequences after a length-1000 gap of noise inputs. Copying a length-1000 sequence requires storing much more information than does waiting 1000 steps to copy a length-10 sequence (to see this, note that only 10 bits need to be stored to copy a length-10 bit sequence, but 1000 bits need to be stored to copy a length-1000 bit sequence. Counting the number of steps requires a logarithmic, not linear number of bits so it’s strictly easier).
> > > >
> > > > Furthermore, rather than training with a curriculum and bumping the sequence length when some proportion of predictions are 100% correct, they always train an LSTM on length ~1000 sequences and report the error in terms of negative-log-likelihood.
> > > >
> > > > > SnAp seems to be learning temporal patterns, however, can you explain why are these long-term dependencies?
> > > >
> > > > I think this is a very insightful point: what is long-term? Upon reflection, we agree that what counts as “long” is quite subjective, as 128 steps really isn’t so long for some datasets but long for others. We have replaced some "long-term" phrasing when talking about temporal dependencies.
> > > >
> > > > As we mention in the paper, one direction we are quite excited about is trying out SnAp with more recent architectures involving self-attention or sparsely accessed memory (e.g. [2, 4]) that are better at taking advantage of long contexts [3, Figure 7] (c.f. [4, Fig 4 (b)]). We have left that for future work.
> > > >
> > > > [1] Mujika et al. Approximating real-time recurrent learning with random kronecker factors. NIPS '18, pp. 6594–6603.
> > > >
> > > > [2] Graves et al. Neural Turing machines. Preprint at http://arxiv.org/abs/1410.5401 (2014).
> > > >
> > > > [3] Jared Kaplan et al. Scaling laws for neural language models, 2020. https://arxiv.org/abs/2001.08361
> > > >
> > > > [4] Jack W Rae et al. Scaling memory-augmented neural networks with sparse reads and writes. NIPS’16, pp. 3628–3636.

---

> > > > > ### Comment · AnonReviewer4 · 2020-11-24
> > > > > **Thank you**
> > > > >
> > > > > Thanks for answering my follow up questions and the latest changes to the manuscript.
> > > > > I modified my score to support accepting your work.

---

### Official Review · AnonReviewer1 · 2020-10-28
**Nice paper; experiment protocol needs improvement**

**Rating:** 7
**Confidence:** 4

**Review:**

## Post response update
The author's response has clarified most of the missing details in the paper. I still have an issue with reporting results for 3 runs --- even if the variance is small for 3 runs, that does not imply that there won't be outliers when one does more runs. Nonetheless, the proposed method is insightful and the paper has significant pedagogical value. I'm moving my score from 6 to 7 and I hope the paper is accepted.

## Summary
The paper tackles the structural credit-assignment problem for a recurrent network when parameter values in earlier time-steps can impact the prediction in the future. The most common approach to achieve this structural credit-assignment, in the current deep learning literature, is BPTT. BPTT, however, is not suitable for online learning. First, the computation and memory requirements of BPTT grow with the length of the sequence. Second, BPTT does not spread computation uniformly --- all the computation happens at the end of an episode. This is not suitable for an online learner that has to learn and react in real-time. An alternative to BPTT is RTRL. The computation and memory needs of RTRL is distributed uniformly across time-steps and RTRL makes it possible to learn at every step, but it requires an intractable amount of memory and compute for even modestly sized networks.

The limitation of both BPTT and RTRL necessities research on new algorithms. Ideally, we want algorithms that spread the computation uniformly and are still scalable. SnAp, the methods proposed in this paper, is one such algorithm. The general idea behind SnAp is to approximate the gradient by only taking into account the impact of a parameter $w_j$ on an activation $h_i$ only if $w_j$ influences $h_i$ with-in n steps. A similar algorithm was used in the original LSTM paper that was equivalent to SnAp with n=1 for the LSTM architecture. This paper generalizes the algorithm used in the original LSTM paper in two dimensions. First, it generalizes it to methods beyond the specific LSTM architecture, and second, it can keep track of influence of parameters across $n$ steps instead of just 1. While the cost of SnAp increases quickly as $n$ increases, the authors propose a promising direction for keeping the cost down. They argue and show that for highly sparse RNNs, SnAp can be scaled to $n > 1$.
## Review

### Strengths
The paper is well written. It summarizes the prior work concisely and explains the two views of computing the gradient for an RNN --- the recursive view used by RTRL and the unrolling view used by BPTT --- clearly. The need for sparsity in RNNs is well-motivated and the observation that sparsity in RNN slows down the propagation of influence of a parameter on a state is interesting. The new algorithm, SnAp, is clearly presented as an approximation to RTRL. The paper also does not make unsubstantiated claims and explains the merits and limitations of the proposed method clearly. Overall, I'm highly impressed by the quality of the paper and the merits of the idea.


### Weaknesses
While the paper excels in writing quality and the proposed method is sound, the empirical evaluation of the method has several issues. First, it's not clear how the hyper-parameters for all the methods were tuned. Were the parameters tuned for SnAp and inherited for other methods? Were they tuned independently? The paper mentions that it used $\beta_1=0.9$ and $\beta_2=0.999$ for the Adam Optimizer without explaining how they were chosen.

Many details of the experiment setup are not fully specified. For example, in page 6 the authors mentioned that they use a one-layer MLP to get 1024 hidden units which are mapped to 256 unit software, but do not clarify if a non-linearity is applied to the 1024 units. The paper also omits how $\theta$ is initialized.

The experimental results have no error margins, and are the mean of only 3 runs. Ideally, authors should repeat the experiments for over 20 runs and report the standard error of the mean. Even if they are limited by available compute and are unable to do many runs, they should at-least report the standard-error for however runs they do (Note that standard error computation is biased for few runs and it might be a good idea to apply bias correction. More details here: https://en.wikipedia.org/wiki/Standard_error).

Parameters sweeps should be extended if the optimal parameter is at the edge. The authors report that they tried $10^{-3}, 10^{-3.5},$ and $10^{-4}$ and found $10^{-3}$ to be best. However, all this tells me is that a higher learning rate could have been even better. They should include larger learning rates in the sweep for the experiment.

It's not clear if the authors are (1) finding the best learning rate and then re-running for 3 seeds for the best learning rate or (2) simply running the experiments for 3 different seeds for all learning rates and reporting the best results. The former is a sound strategy whereas the latter suffers from over-estimation bias.

Given the issues in the experiment methodology, I'm giving the paper a weak accept for now even though I think that the paper excels in many ways. The issues identified can easily be fixed during the discussion period and I would be more than happy to change my score to an accept or a strong accept after a revision that fixes the experimental issues.


### Questions
1. In page two, the paper introduces a function $g_{\phi}$ for mapping the state to the target and then says that the goal of the learner is to minimize loss with respect to $\theta.$ Shouldn't the loss be minimized with respect to both $\theta$ and $\phi$? Or are the authors implying that the readout layer is fixed? (Or perhaps $\phi$ is a subset of $\theta$?)

2. It's not clear to me from the write-up how the sparsity pattern is choosen empirically. Is the idea to run the RNN for n-steps, empirically observe enteries in $J_n$ that are zero, and fix those enteries to be zero for all future steps? If yes, could the initial weights of the RNN and the data make an entry in $J_n$ to be zero even if it would not have been zero for a different initialization and data-stream?

---

> ### Author Response · Authors · 2020-11-17
> **Official Reply**
>
> We thank reviewer #1 for their helpful review.
>
> The Adam parameters are the default of the framework we used and were not tuned for any of the techniques.  There is a ReLU non-linearity in between the 1024-unit MLP and the 256-unit softmax.  $\theta$ is initialized using a truncated normal distribution whose std. deviation is the inverse of the square root of the fan-in.  The embedding matrix is not shared between the input and the output.
>
> We note that the plots for the copy task (figure 4) do show the variance across 3 runs (the variance is quite small for many of the runs and barely visible).
>
> This learning rate sweep came from [1].  We have now extended it to also include $10^{-2.5}$ and still find that $10^{-3}$ is the best in almost all cases. Higher-orders of SnAp (SnAp-2 and SnAp-3) perform better with this larger learning rate for a very small number of configurations, but for simplicity and at a slight disadvantage to SnAp we will continue to report results for all methods with $10^{-3}$. The differences in performance between the methods are large and we do not believe that methodology (1) or (2) would lead to any different conclusions.
>
> Currently we use the simple strategy of picking a sparsity pattern by selecting edges uniformly at random.  There are more sophisticated strategies that are not immediately compatible with RTRL, such as RigL [2], but we believe that combining these sparse training techniques with RTRL is future work.
>
> > In page two, the paper introduces a function $g_{\phi}$ for mapping the state to the target and then says that the goal of the learner is to minimize loss with respect to $\theta.$ Shouldn't the loss be minimized with respect to both $\theta$ and $\phi$? Or are the authors implying that the readout layer is fixed? (Or perhaps $\phi$ is a subset of $\theta$?)
>
> We agree it is slightly confusing that the exposition does not comment on the learning of readout parameters $\phi$. We do train the parameters of the readout, but we separated them out for the purposes of the exposition because the core issue is gradient computation for parameters which affect recurrent state. The gradients w.r.t $\phi$ can always be accumulated forward in time without special consideration for BPTT versus RTRL. As you can see the code snippet in our reply to reviewer 2, we use backpropagation to compute the readout gradients and can simply forget any readout activations used in previous timesteps without any approximation being made.
>
> [1] Asier Mujika, Florian Meier, and Angelika Steger. Approximating real-time recurrent learning with random kronecker factors. In S. Bengio, H. Wallach, H. Larochelle, K. Grauman, N. Cesa-Bianchi, and R. Garnett (eds.), Advances in Neural Information Processing Systems 31, pp. 6594–6603. 2018.
>
> [2] Evci, U., Gale, T., Menick, J., Castro, P. S., and Elsen, E. (2020). Rigging the lottery: Making all tickets winners. In Proceedings of the 37th International Conference on Machine Learning.

---

### Official Review · AnonReviewer3 · 2020-10-28
**iterative work on adding sparsity to RNNs, need more clarification**

**Rating:** 7
**Confidence:** 3

**Review:**

## Second Review
The author's thoughtful response has clarified most of the missing details in the paper. It is true that idea is interesting and theoretical analysis are promising. However, I still have issues understanding failure conditions. If the method is purely based on trial and error to determine optimal threshold for sparsity, then it requires many engineering tricks. Thus I would request authors to provide such details, such that young researchers can extend this work to create better training paradigm for RNNs. Other issue as pointed out by other reviewers is using only 3 runs to report results.  I really appreciate explanation about difference between 2 variants of GRU and how it adds sparsity to the model. Additionally JAX code helped in understanding many key apsects of the work. I would encourage authors to make it public, and provide key insights for training RNNs using snap. Nonetheless, the proposed method and theoretical analysis are insightful and I believe this to be a first step towards building scalable RNNs which efficiently gets rid of credit assignment issue. This paper does add significant pedagogical value, which can benefit complex task such as grammatical inference. I'm increasing my score from 5 to 7 and I hope this paper is accepted.

## Summary
Paper introduces snap which adds sparsity to the influence matrix extracted from RTRL which acts as a practical approximation for RTRL, snap is extension of prior work on snap-1 used to train LSTM, and authors have shown that one can train dense as well as sparse RNNs using snap achieving similar performance as BPTT on real as well as synthetic dataset. Few clarifications in terms of snap working and few key information w.r.t to parameters are missing.

## Clarification

How does one evaluate level of sparsity required for any given task? At what n (sparsity ratio) optimal performance is observed which leads to better performance. It is well known that full RTRL (forward propagation helps compared with backward propagation), especially in case of continual or online learning [Ororbia and Mali 2020] and copy task (KF-RTRL, UORO). Does current sparsity measure work on variant of RTRL? And how do you determine top k to select k elements for creating a sparse matrix? is it random like 70-80-99? Many key details are missing, and authors are requested to provide more information to better understand model flow.

In appendix authors talk about “modeling performance of the two variants of GRU. which has been shown to be largely the same, but the second variant is faster and results in sparser D_t and I_t”. I am confused, what is the relationship between sparsity and two variants? Please provide some numbers explaining how sparsity is increased by moving reset gate after the matrix multiplication.

How does sparsity measure is introduced in this work? Does model stay consistent whenever regularization approaches such as zoneout or dropout are used or introduced into the model? Do you observe the similar performance? Does network roughly converge to similar performance with optimal sparsity or sparsity measure changes as other regularization approaches are introduced? Did you do grid search for language modelling task or copy task (beside learning rate)? If so please provide details? Citation and comparison missing with Sparse attentive backtracking, which in theory can work with sparse networks and its temporal credit assignment mechanism can help in introducing sparsity [ke and goyal 2019].

Authors states that “In order to induce sparsity, we generate a sparsity pattern uniformly at random and fix it throughout training” What is the range for random uniform? Is model sensitive whenever sparsity pattern is changed while training (may be per epoch or k epochs). How can one ensure that the sparsity pattern at start is the optimal one for any network? Does similar pattern work for all GRU, LSTM, RNN or one needs to adapt scheme based on architecture?

Advantage of snap-2 and 3 over snap-1, snap-1 is similar to (Hochreiter & Schmidhuber, 97) work on training LSTM, what modification is introduced on snap-1 beside training it on GRU? And sparse networks. It is still unclear what advantage these 3 variants add. It is important to show speed (with various sparsity, convergence plots or else these variants would have similar performance and memory requirement compared with vanilla RTRL


[Ke and Goyal 2018] Ke, N.R., GOYAL, A.G.A.P., Bilaniuk, O., Binas, J., Mozer, M.C., Pal, C. and Bengio, Y., 2018. Sparse attentive backtracking: Temporal credit assignment through reminding. In Advances in neural information processing systems (pp. 7640-7651).

[Ororbia and Mali 2020] Ororbia, A., Mali, A., Giles, C.L. and Kifer, D., 2020. Continual learning of recurrent neural networks by locally aligning distributed representations. IEEE Transactions on Neural Networks and Learning Systems

---

> ### Author Response · Authors · 2020-11-17
> **Official Reply**
>
> We thank the reviewer for their comments and want to help clarify some of the confusion.
>
> Changing the size of a network changes the capacity of a network.  Choosing the right size of a dense network for a given task is often a matter of trial and error.  Sparsity also changes the capacity of a model, and the right combination of size and sparsity generally must also be determined with trial and error.  Previous work, for example, Efficient Neural Audio Synthesis, has shown that for RNNs efficiency increases as the sparsity level increases, up to at least ~99%, which is also what we observe on language modeling in this paper.
>
> > In appendix authors talk about “modeling performance of the two variants of GRU….what is the relationship between sparsity and two variants?
>
> The key is that variant 2 avoids “the composition of parameterized linear maps within a single RNN step”.  If we remove the non-linearities for clarity we can see that variant 1 involves a product of two parameter matrices W_ha and W_ir.  In variant 2, there is no matrix product of parameter matrices.  This is the key difference.  And note that this specifically refers to the sparsity of the jacobian of the state w.r.t. the parameters.
>
> > How does sparsity measure is introduced in this work? Does model stay consistent whenever regularization approaches such as zoneout or dropout are used or introduced into the model?
>
> We have not experimented with other forms of regularization like dropout and zoneout because very sparse networks are already regularized due to the removal of a large proportion of their parameters. We encourage the reviewer to see Figure 4 where we try many different sparsity levels for a few different RNN variants.  Investigating the combination of dropout or zoneout in conjunction with sparse parameters seems like an interesting idea, but is orthogonal to focus of the paper on RTRL.
>
> > Authors states that “In order to induce sparsity, we generate a sparsity pattern uniformly at random and fix it throughout training” What is the range for random uniform?...
>
> We agree this could be made more explicit and have done so in the updated draft. When we say random uniform, we mean a uniform choice over which (set of) parameters are removed. Alternatively, we could say that an independent bernoulli choice is made for each parameter, whether it should be kept or removed (forced to zero).  The weights themselves are initialized from the same distribution that would be used for a dense model.
>
> > what modification is introduced on snap-1 beside training it on GRU?
>
> As discussed in Section 4 (Related Work), We have defined SnAp-1 in general terms for any recurrent architecture, but it ends up being quite similar to the algorithm used to train LSTM in the original Hochreiter & Schmidhuber 1997 paper. The exact details of that algorithm are pretty hard to discern from the paper’s exposition but it’s clearly similar in flavour (tracking the influence of a parameter within a small subset of units). We have explained our version of the idea simply and formally, characterised its computational properties versus alternative algorithms in the literature, extended the idea to any recurrent architecture and generalized it to an n-step instead of 1-step approximation.
>
> > It is important to show speed (with various sparsity, convergence plots or else these variants would have similar performance and memory requirement compared with vanilla RTRL
>
> We’d refer you to table 2, where FLOPs serve as a proxy for speed and we show FLOPs cost as a multiple of BPTT or RTRL cost.  We note that prior published papers on RTRL approximations have not been competitive on a wall-clock basis with BPTT.  Our implementation takes advantage of some of the memory and compute savings that are possible with SnAp, but not all of them.  Current popular deep learning frameworks make it difficult to take full advantage of sparse linear algebra, but we hope that work such as ours will spur development in this area.

---

### Official Review · AnonReviewer2 · 2020-10-28
**A promising study (second review: even better than I thought!)**

**Rating:** 8
**Confidence:** 5

**Review:**

## Second review

Thanks for taking all my comments seriously. After clarification of the difference with RFLO I see that this work is even richer than I thought and I increase my grade to 8. It seems that other reviewers did not appreciate that training a network without back-prop requires nontrivial engineering and theoretical considerations which are well described in this paper, I truly think it is a pity if this work is not accepted.

I fully agree with the difference between RFLO and Snap-1 that you describe in your reply, and I think it would be really great to put that somewhere in the paper. As you suggest it would be great to explain that you did not use random feedback weights for RFLO.

This would also be a great opportunity to explain how did you extend RFLO to a GRU network in Figure 3. I find it a bit puzzling, that RFLO appears worse than an untrained network in Figure 3 (even early in training as in seen in Figure 3.B). Is there any additional difference in the network model for these two baselines, like one is using leaky RNN and the other one GRU or something like that?

I find the piece of JAX code incredibly rich. It would be great to publish that along with the paper! JAX is not yet very well spread, and we see here that it is a very promising tool for custom gradient in RNNs.

## Summary

The authors describe new algorithms to train sparse recurrent neural networks, these algorithms are described as variants of RTRL. These methods, called SNAP-$n$, use the same induction as in RTRL but approximate the true Jacobian matrix $J$ by a sparse matrix where each coefficient is set to zero if the corresponding parameter does not influence the corresponding state variable within $n$ steps. These alternatives to BPTT alleviate the memory consumption growing otherwise linearly with the sequence length.

A theoretical complexity analysis and simulation experiments are carried out. The simulations are performed on the character prediction task and a synthetic copy task. The authors report that the network reaches performance comparable to BPTT and sometimes better (snap-3 leads to better copy task performance with GRU, snap-2 seems already better with LSTMs, however it requires many more FLOPS).

##  General opinion

Congratulations to the authors. This is an important topic since recurrent networks are not efficiently trained with BPTT. More intensive and rigorous research are needed to find suitable alternatives. The SNAP idea is simple and appealing, and the results encouraging (even though it still requires a large number of FLOPS).

I recomputed rather carefully the complexity analysis for sparse RTRL, snap 1 and snap 2 and arrived at similar results. Theoretical results seem to be correct and the experiments are credible.

##  Requires clarification

One negative comment is that snap-1 has been published before in other forms as explained below The writing makes it sounds more novel that it actually is and some comparisons with other algorithms are unfair. It would be great to correct the shot but snap deserves publication anyway since it also provides a rigorous analysis of the full snap family and snap-1 had not been explored in such details in the interesting context of sparse networks.

RFLO for leaky RNNs is basically exactly snap-1 with the additional burden of carrying random feedbacks. It is written three times that snap-1 is better than RFLO but it would be great to comment on the differences and explain where is the difference of performance coming form. The random feedback was introduced in RFLO to avoid the transport problem for biological plausibility. It does not seem relevant to do this extra approximation step if RFLO is used for performance and not plausibility. If the random feedback is the main difference, one should clearly say that the algorithm are otherwise identical. If the authors see other differences, it would be great to indicate what they are.

This very same algorithm (snap-1, RFLO) was also published under the name e-prop 1 [a] although the theory was derived differently. The authors had shown that e-prop 1 (aka snap-1) works well on the copy task, a word prediction task with LSTMs (Figure 4 in [a]), ATARI games and TIMIT (more recent work).  The authors had also suggested an amelioration called e-prop 3 that improved the performance and kept the same time complexity as BPTT unlike snap-2 and snap-3. Maybe it is relevant to comment on the relationship between snap and this paper [a] ?

In case the authors were not aware of this, an other interesting approximation to RTRL was published in [b], the authors may or may not comment about that too.

There are technical details that should be given for clarification:
- The authors might want to provide details on the computation of the complexity of one or two of the essential component of the table to make it more accessible to the reader. If I am not mistaken I think that the complexity results are true "up-to a proportionality constant" for a general RNN model, maybe it would be great to write that in the caption for instance. The complexity is written in the table in the form A + B, I understood than A is the complexity in the forward pass and B is the complexity in the backward pass, but maybe it should be explained. Maybe there is also an easy way to confirm those numbers with the empirical number of FLOPS given later in the paper ?

- In Figure 4, I cannot find out what the colored dash lines are meant to be. This caption is rather short and there is an opportunity to add some information: for instance what is "curricula max".

- In Figure 2, the author use k^2 by they have also introduce the letter p. Is that not meant to be the same thing? It probably depends on the RNN model? Before doing the calculation of the complexity myself I did not know whether p included already the coefficient d, I do not think that it obvious that zeroed coefficients are still considered as "parameters" in the "number of parameters" p. Maybe this can be said in the caption?

- If I understand correctly the pruning method in appendix B, the network is first trained until convergence with BPTT and then, the best architecture is fixed to be retrained with snap? It would be great to clarify this paragraph because it is not easy to read. If that's the case there could be a substantial transfer of information by passing on the "winning" architecture from BPTT to the SNAP training, in particular for very sparse networks. Is that really necessary? How much would the performance decrease? As a control, I would guess that for something like 75% sparsity the network can be trained from a random matrix. Also I do think that there are now simple methods available for training much sparser networks from scratch, and it would not require this pre-BPTT step.

- It would be so great to have some details about the implementation: what software did you use to perform this "remake" of a forward propagation? Did you have to implement a custom C++/cuda code, maybe use JAX ? Did you use sparse matrix cuda kernel, how good was it? Maybe the code will be shared, if not any details are welcome.


[a] Biologically inspired alternatives to backpropagation through time for learning in recurrent neural nets
Guillaume Bellec, Franz Scherr, Elias Hajek, Darjan Salaj, Robert Legenstein, Wolfgang Maass
https://arxiv.org/abs/1901.09049

[b] Kernel RNN Learning (KeRNL)
Christopher Roth, Ingmar Kanitscheider, Ila Fiete
https://openreview.net/forum?id=ryGfnoC5KQ

---

> ### Author Response · Authors · 2020-11-17
> **Official Reply**
>
> We thank reviewer #2 for their very thorough review.
>
> Thanks for making it clear that we did not describe the differences between SnAp-1 and RFLO clearly enough.  First, we would like to note that we did not use random feedback in our implementation of RFLO: we used exact feedback with the structure of the updates proposed by RFLO.  We will clarify this in the paper.
>
> We believe that even in the case of leaky RNNs RFLO and SnAp-1 are different.  To show this we take equation 22 from section 3.6 of [1], and rewrite it in the notation used in our paper in equation 3 (which we also rewrite for a leaky RNN).  Here $i$ is defined as $i=u(j)$ as it is in our paper.
>
> SnAp-1 = $(J_t)_\{ij\} = \alpha(I_t)_\{ij\} + (\alpha D_t + (1 - \alpha))_\{ii\}(J_\{t-1\})_\{ij\}$
>
> RFLO = $(J_t)_\{ij\} = \alpha(I_t)_\{ij\} + (1 - \alpha)(J_\{t-1\})_\{ij\}$
>
> The key difference being that the previous Jacobian is multiplied only by the constant $(1 - \alpha)$ in the case of RFLO, whereas in SnAp-1 it will have a dependence on $D_t$, even for a leaky RNN. Also - we quote from appendix 1 of the RFLO paper [2] “Second, we simply drop the terms involving 𝐖 in Equation (11), so that nonlocal information about all recurrent weights in the network is no longer required to update a particular synaptic weight.”
>
> This is an important point to get right and we would appreciate the reviewer feedback on this analysis.  Once we are in agreement, we will add this clarification to the paper.
>
> Thank you for the reference to the e-prop paper, it is indeed exciting work.  After reviewing it, we agree that eprop-1 and SnAp-1 are indeed essentially describing the same idea, although with very different expositions.  It is unfortunate that the -1 postfix has a different meaning in the two names.  We will update the paper appropriately to reflect this prior work and emphasize our different exposition.
>
> Eprop-3 is a clever way to combine RTRL and synthetic gradients with BPTT.  RTRL allows for information to be carried from the past into the truncation window and synthetic gradients allow for a better estimate of the gradient from ahead of the truncation window.  Using eprop-1 / SnAp-1 makes it possible to compute the RTRL component with a similar time complexity to BPTT.  Eprop-3 seems well suited to the not fully online setting, whereas Snap-2/3 are still suitable in a fully online setting.  In the offline setting it might be interesting to consider using SnAp-2 to replace eprop-1 / SnAp-1 to allow for more information carried from the past.
>
> Addressing technical details:
>
> We will add some details to the appendix describing how these values are obtained.  The form A + B is indeed where A is the cost of going forward and B is the cost of the backward calculation in BPTT or the influence matrix update in RTRL variants.
> We will fix the figure to make this clearer.  The solid lines are when updates are done online (i.e. T = 1) and the dashed lines are when updates are only done at the end of each episode.
>
> Yes, we distinguish between $k^2$ and $p$ because while for a vanilla RNN, they are the same, they might not be in architectures such as GRU and LSTM.  It is confusing that $p$ refers to always the dense parameter size here, we will make this explicit.
> In this section in the appendix we use weight magnitude during training with BPTT, there is no retraining with SnAp.
>
> This example is meant to be motivating - but achieving this level of performance with SnAp would also require a sparse training method such as RigL [3] that is compatible with SnAp, which we believe is beyond the scope of this paper.
>
> We used JAX and indeed we found it quite nice for this type of research. Whilst we can’t do a full code release at this time, we can attach a working, simplified snippet showing how we implemented SnAp with JAX. https://drive.google.com/file/d/1OcGSdsfKkIg9uibqblRwGJ5E5NxEdES4/view?usp=sharing
>
>
> [1] Owen Marschall and Kyunghyun Cho and Cristina Savin, “A Unified Framework of Online Learning Algorithms for Training Recurrent Neural Networks”, Journal of Machine Learning Research, 2020, http://jmlr.org/papers/v21/19-562.html
>
> [2] James M Murray. Local online learning in recurrent networks with random feedback. eLife, 8:e43299, 2019.
>
> [3] Evci, U., Gale, T., Menick, J., Castro, P. S., and Elsen, E. (2020). Rigging the lottery: Making all tickets winners. In Proceedings of the 37th International Conference on Machine Learning.

---

### Decision · Program_Chairs · 2021-01-07
**Final Decision**

**Decision:**

Accept (Spotlight)

**Comment:**

This paper introduces a method for approximating real-time recurrent learning (RTRL) in a more computationally efficient manner. Using a sparse approximation of the Jacobian, the authors show how they can reduce the computational costs of RTRL applied to sparse recurrent networks in a manner that introduces some bias, but which manages to preserve good performance on a variety of tasks.

The reviewers all agreed that the paper was interesting, and all four reviewers provided very thorough reviews with constructive criticisms. The authors made a very strong effort to attend to all of the reviewers' comments, and as a result, some scores were adjusted upward. By the end, all reviewers had provided scores above the acceptance threshold.

In the AC's opinion, this paper is of real interest to the community and may help to develop new approaches to training RNNs at large-scale. As such, the AC believes that it should be accepted and considered for a spotlight.